# DCMIP2016: The Splitting Supercell Test Case

Colin M. Zarzycki[1,2], Christiane Jablonowski[3], James Kent[4], Peter H. Lauritzen[2], Ramachandran Nair[2], Kevin A. Reed[5], Paul A. Ullrich[6], David M. Hall[7,8], Mark A. Taylor[9], Don Dazlich[10], Ross Heikes[10], Celal Konor[10], David Randall[10], Xi Chen[11], Lucas Harris[11], Marco Giorgetta[12], Daniel Reinert[13], Christian Kühnlein[14], Robert Walko[15], Vivian Lee[16], Abdessamad Qaddouri[16], Monique Tanguay[16], Hiroaki Miura[17], Tomoki Ohno[18], Ryuji Yoshida[19], Sang-Hun Park[20], Joseph Klemp[2], and William Skamarock[2]

[1]Pennsylvania State University
[2]National Center for Atmospheric Research
[3]University of Michigan
[4]University of South Wales
[5]Stony Brook University
[6]University of California, Davis
[7]University of Colorado, Boulder
[8]NVIDIA Corporation
[9]Sandia National Laboratories
[10]Colorado State University
[11]Geophysical Fluid Dynamics Laboratory (GFDL)
[12]Max Planck Institute for Meteorology
[13]Deutscher Wetterdienst (DWD)
[14]European Centre for Medium-Range Weather Forecasts (ECMWF)
[15]University of Miami
[16]Environment and Climate Change Canada (ECCC)
[17]University of Tokyo
[18]Japan Agency for Marine-Earth Science and Technology
[19]RIKEN
[20]Yonsei University

*Correspondence to:* Colin Zarzycki (czarzycki@psu.edu)

**Abstract.** This paper describes the splitting supercell idealized test case used in the 2016 Dynamical Core Model Intercomparison Project (DCMIP2016). These storms are useful testbeds for global atmospheric models because the horizontal scale of convective plumes is O(1km), emphasizing non-hydrostatic dynamics. The test case simulates a supercell on a reduced radius sphere with nominal resolutions ranging from 4km to 0.5km and is based on the work of Klemp et al. (2015). Models are initialized with an atmospheric environment conducive to supercell formation and forced with a small thermal perturbation. A simplified Kessler microphysics scheme is coupled to the dynamical core to represent moist processes. Reference solutions for DCMIP2016 models are presented. Storm evolution is broadly similar between models, although differences in final solution exist. These differences are hypothesized to result from different numerical discretizations, physics-dynamics coupling, and numerical diffusion. Intramodel solutions generally converge as models approach 0.5km resolution, although exploratory simulations at 0.25km imply some dynamical cores require more refinement to fully converge. These results can be used as a

reference for future dynamical core evaluation, particularly with the development of non-hydrostatic global models intended to be used in convective-permitting regimes.

## 1 Introduction

Supercells are strong, long-lived convective cells containing deep, persistent rotating updrafts that operate on spatial scales O(10km). They can persist for many hours and frequently produce large hail, tornados, damaging straight line winds, cloud-to-ground lightning, and heavy rain (Browning, 1964; Lemon and Doswell, 1979; Doswell and Burgess, 1993). Therefore, accurate simulation of these features is of great societal interest and critical for atmospheric models.

The supercell test applied in DCMIP2016 (Ullrich et al., 2017) permits the study of a non-hydrostatic moist flow field with strong vertical velocities and associated precipitation. This test is based on the work of Klemp and Wilhelmson (1978) and Klemp et al. (2015) and assesses the performance of global numerical models at extremely high spatial resolution. It has recently been used in the evaluation of next-generation weather prediction systems (Ji and Toepfer, 2016).

Previous work regarding the role of model numerics in simulating extreme weather features has generally focused on limited area domains (e.g., Gallus and Bresch (2006); Guimond et al. (2016)). While some recent work has targeted global frameworks and extremes – primarily tropical cyclones (e.g., Zhao et al. (2012); Reed et al. (2015)) – these studies have almost exclusively employed hydrostatic dynamical cores at grid spacings approximately $0.25°$ and coarser.

The supercell test here emphasizes resolved, non-hydrostatic dynamics. In this regime the effective grid spacing is very similar to the horizontal scale of convective plumes. Further, the addition of simplified moist physics injects energy near the grid-scale in a conditionally-unstable atmosphere, which imposes significant stress on model numerics. The supercell test case therefore sheds light on the interplay of the dynamical core and subgrid parameterizations and highlights the impact of both implicit and explicit numerical diffusion on model solutions. It also demonstrates credibility of a global modeling framework to simulate extreme phenomena, essential for future weather and climate simulations.

## 2 Description of test

The test case is defined as follows. The setup employs a non-rotating reduced-radius sphere with scaling factor $X = 120$. Reducing the model's planetary radius allows for fine horizontal grid spacing and non-hydrostatic motions to be resolved at relatively low computational cost compared to a configuration using the actual size of the Earth (Kuang et al., 2005). Wedi and Smolarkiewicz (2009) provide a detailed overview of the reduced-radius framework for testing global models. For a $1°$ mesh, the grid spacing of the reduced radius sphere is approximately $1°/X \sim 111\text{km}/X \sim 111\text{km}/120 \sim 1\text{km}$ near the equator. Klemp et al. (2015) demonstrated excellent agreement between simulations using this value of $X$ and those completed on a flat, Cartesian plane with equivalent resolution. The model top ($z_t$) is placed at 20km with uniform vertical grid spacing ($\Delta z$) equal to 500m, resulting in 40 full vertical levels. No surface drag is imposed at the lower boundary (free slip condition). Water vapor ($q_v$), cloud water ($q_c$) and rain water ($q_r$) are handled by a simple Kessler microphysics routine (Kessler, 1969). In particular,

the Kessler microphysics used here is outlined in detail in Appendix C of Klemp et al. (2015) and code for reproducing this configuration is available via the DCMIP2016 repository (http://dx.doi.org/10.5281/zenodo.1298671).

All simulations are integrated for 120min. Outputs of the full three-dimensional prognostic fields as well as all variables pertaining to the microphysical routines were stored for post-processing at least every 15min. Four different horizontal resolutions were specified; $4°$, $2°$, $1°$, and $0.5°$. For the reduced-radius framework, this results in approximate grid spacings of 4km, 2km, 1km, and 0.5km, respectively. Note that here we use '(nominal) resolution' and 'grid spacing' interchangeably to refer to the horizontal length of a single grid cell or distance between gridpoints. All relevant constants mentioned here and in the following section are defined in Table 1.

## 2.1  Mean atmospheric background

The mean atmospheric state is designed such that it consists of large instability (convective available potential energy (CAPE) of approximately 2200 $m^2$ $s^{-1}$) and strong low-level wind shear, both of which are strong precursors of supercell formation (Weisman and Klemp, 1982).

The definition of this test case relies on hydrostatic and cyclostrophic wind balance, written in terms of Exner pressure $\pi$ and virtual potential temperature $\theta_v$ as

$$\frac{\partial \pi}{\partial z} = -\frac{g}{c_p \theta_v}, \quad \text{and} \quad u^2 \tan\varphi = -c_p \theta_v \frac{\partial \pi}{\partial \varphi}. \tag{1}$$

Defining $u = u_{eq} \cos\varphi$ to maintain solid body rotation, where $u_{eq}$ is the equatorial wind velocity, these equations can be combined to eliminate $\pi$, leading to

$$\frac{\partial \theta_v}{\partial \varphi} = \frac{\sin(2\varphi)}{2g} \left( u_{eq}^2 \frac{\partial \theta_v}{\partial z} - \theta_v \frac{\partial u_{eq}^2}{\partial z} \right). \tag{2}$$

The wind velocity is analytically defined throughout the domain. Meridional and vertical wind is initially set to zero. The zonal wind is obtained from

$$u(\varphi, z) = \begin{cases} \left( U_s \dfrac{z}{z_s} - U_c \right) \cos(\varphi) & \text{for } z < z_s - \Delta z_u, \\[2ex] \left[ \left( -\dfrac{4}{5} + 3\dfrac{z}{z_s} - \dfrac{5}{4}\dfrac{z^2}{z_s^2} \right) U_s - U_c \right] \cos(\varphi) & \text{for } |z - z_s| \leq \Delta z_u \\[2ex] (U_s - U_c) \cos(\varphi) & \text{for } z > z_s + \Delta z_u \end{cases} \tag{3}$$

The equatorial profile is determined through numerical iteration. Potential temperature at the equator is specified via

$$\theta_{\text{eq}}(z) = \begin{cases} \theta_0 + (\theta_{tr} - \theta_0) \left( \dfrac{z}{z_{tr}} \right)^{\frac{5}{4}} & \text{for } 0 \leq z \leq z_{tr}, \\[2ex] \theta_{tr} \exp\left( \dfrac{g(z - z_{tr})}{c_p T_{tr}} \right) & \text{for } z_{tr} \leq z \end{cases} \tag{4}$$

And relative humidity is given by

$$
H(z) = \begin{cases} 1 - \dfrac{3}{4}\left(\dfrac{z}{z_{tr}}\right)^{5/4} & \text{for } 0 \leq z \leq z_{tr}, \\[2ex] \dfrac{1}{4} & \text{for } z_{tr} \leq z. \end{cases}
\tag{5}
$$

It is assumed that the saturation mixing ratio is given by

$$
q_{vs}(p,T) = \left(\frac{380.0}{p}\right)\exp\left(17.27 \times \frac{T-273.0}{T-36.0}\right)
\tag{6}
$$

Pressure and temperature at the equator are obtained by iterating on hydrostatic balance with initial state

$$
\theta_{v,\mathrm{eq}}^{(0)}(z) = \theta_{\mathrm{eq}}(z),
\tag{7}
$$

and iteration procedure

$$
\pi_{\mathrm{eq}}^{(i)} = 1 - \int_0^z \frac{g}{c_p \theta_{v,\mathrm{eq}}^{(i)}}\,dz
\tag{8}
$$

$$
p_{\mathrm{eq}}^{(i)} = p_0(\pi_{\mathrm{eq}}^{(i)})^{c_p/R_d}
\tag{9}
$$

$$
T_{\mathrm{eq}}^{(i)} = \theta_{\mathrm{eq}}(z)\pi_{\mathrm{eq}}^{(i)}
\tag{10}
$$

$$
q_{\mathrm{eq}}^{(i)} = H(z)q_{vs}(p_{\mathrm{eq}}^{(i)}, T_{\mathrm{eq}}^{(i)})
\tag{11}
$$

$$
\theta_{v,\mathrm{eq}}^{(i+1)} = \theta_{\mathrm{eq}}(z)(1 + M_v q_{\mathrm{eq}}^{(i)})
\tag{12}
$$

This iteration procedure generally converge to machine epsilon after approximately 10 iterations. The equatorial moisture profile is then extended through the entire domain,

$$
q(z,\varphi) = q_{\mathrm{eq}}(z).
\tag{13}
$$

Once the equatorial profile has been constructed, the virtual potential temperature through the remainder of the domain can be computed by iterating on (2),

$$
\theta_v^{(i+1)}(z,\varphi) = \theta_{v,\mathrm{eq}}(z) + \int_0^\varphi \frac{\sin(2\phi)}{2g}\left(u_{eq}^2\frac{\partial\theta_v^{(i)}}{\partial z} - \theta_v^{(i)}\frac{\partial u_{eq}^2}{\partial z}\right)d\varphi.
\tag{14}
$$

Again, approximately 10 iterations are needed for convergence to machine epsilon. Once virtual potential temperature has been computed throughout the domain, Exner pressure throughout the domain can be obtained from (1),

$$
\pi(z,\varphi) = \pi_{eq}(z) - \int_0^\varphi \frac{u^2\tan\varphi}{c_p\theta_v}\,d\varphi,
\tag{15}
$$

and so

$$
p(z,\varphi) = p_0\pi(z,\varphi)^{c_p/R_d},
\tag{16}
$$

$$
T_v(z,\varphi) = \theta_v(z,\varphi)(p/p_0)^{R_d/c_p}.
\tag{17}
$$

Note that, for (13-14), Smolarkiewicz et al. (2017) also derive an analytic solution for the meridional variation of the initial background state for shallow atmospheres.

## 2.2 Potential temperature perturbation

To initiate convection, a thermal perturbation is introduced into the initial potential temperature field:

$$
\theta'(\lambda,\phi,z) = \begin{cases} \Delta\theta\cos^2\left(\frac{\pi}{2}R_\theta(\lambda,\varphi,z)\right) & \text{for } R_\theta(\lambda,\varphi,z) < 1, \\[2ex] 0 & \text{for } R_\theta(\lambda,\varphi,z) \geq 1, \end{cases} \tag{18}
$$

where

$$
R_\theta(\lambda,\varphi,z) = \left[\left(\frac{R_c(\lambda,\varphi;\lambda_p,\varphi_p)}{r_p}\right)^2 + \left(\frac{z-z_c}{z_p}\right)^2\right]^{1/2}. \tag{19}
$$

An additional iterative step is then required to bring the potential temperature perturbation into hydrostatic balance. Without this additional iteration, large vertical velocities will be generated as the flow rapidly adjusts to hydrostatic balance since the test does not possess strong non-hydrostatic characteristics at initialization. Plots showing the initial state of the supercell are shown in Figs. 1 and 2 for reference. Code used by modeling centers during DCMIP2016 for initialization of the supercell test case is archived via Zenodo (http://dx.doi.org/10.5281/zenodo.1298671).

The test case is designed such that the thermal perturbation will induce a convective updraft immediately after initialization. As rain water is generated by the microphysics, reduced buoyancy and a subsequent downdraft at the equator in combination with favorable vertical pressure gradients near the peripheral flanks of the storm will cause it to split into two counterrotating cells that propagate transversely away from the equator until the end of the test (Rotunno and Klemp, 1982, 1985; Rotunno, 1993; Klemp et al., 2015).

## 2.3 Physical and Numerical Diffusion

As noted in Klemp et al. (2015), dissipation is an important process near the grid-scale, particularly in simulations investigating convection in unstable environments such as this. To represent this process and facilitate solution convergence as resolution is increased for a given model, a second-order diffusion operator with a constant viscosity (value) is applied to all momentum equations ($\nu = 500 \text{ m}^2 \text{ s}^{-1}$) and scalar equations ($\nu = 1500 \text{ m}^2 \text{ s}^{-1}$). In the vertical, this diffusion is applied to the perturbation from the background state only in order to prevent the initial perturbation from mixing out.

Models that contributed supercell test results at DCMIP2016 are listed in Table 2. They are formally described in Ullrich et al. (2017) and the references therein. Further, specific versions of the code used in DCMIP2016 and access instructions are also listed in Ullrich et al. (2017). Note that not all DCMIP2016 participating groups submitted results for this particular test.

Due to the multitude of differing implicit and explicit diffusion in the participating models, some groups chose to apply variations in how either horizontal or vertical diffusion were treated in this test case. Deviations from the above specified diffusion are as follows. CSU applied uniform three-dimensional second order diffusion with coefficients of $\nu = 1500 \text{ m}^2\text{s}^{-1}$

for $q_v$ and $\theta_v$, $\nu = 1000\,\mathrm{m^2s^{-1}}$ for $q_c$ and $q_r$, and $\nu = 500\,\mathrm{m^2s^{-1}}$ for divergence and relative vorticity. FV³ applied divergence and vorticity damping separately to the velocity fields along the floating Lagrangian surface. A Smagorinsky diffusion is also applied to the horizontal wind. ICON applied constant horizontal second-order diffusion to the horizontal and vertical velocity components ($\nu = 500\,\mathrm{m^2s^{-1}}$) as well as the scalar variables $\theta_v$ and $q_{v,c,r}$ ($\nu = 1500\,\mathrm{m^2s^{-1}}$). No explicit diffusion was applied in the vertical. NICAM applied a dynamically-defined fourth-order diffusion to all variables in the horizontal with vertical dissipation being implicitly handled by the model's vertical discretization.

## 3 Results

The following section describes the results of the supercell test case at DCMIP2016, both from a intermodel time evolution perspective and intramodel sensitivity to model resolution and ensuing convergence. Note that there is no analytic solution for the test case, but features specific to supercells should be observed and are subsequently discussed. It is not the intent of this manuscript to formally explore the precise mechanisms for model spread or define particular solutions as superior, but rather, to publish an overview set of results from a diverse group of global, non-hydrostatic models to be used for future development endeavours. Future work employing this test case in a more narrow sense can isolate some of the model design choices that impact supercell simulations.

### 3.1 Time evolution of supercell at control resolution

Fig. 3 shows the temporal evolution (every 30min, out to 120min test termination) of the supercell for contributing models at the control resolution of 1km. The top four panels for each model highlight a cross-section at 5km elevation through vertical velocity ($w$) while the bottom four show a cross-section (at the same elevation) through the rain water ($q_r$) field produced by the Kessler microphysics. For $w$, red contours represent rising motion while blue contours denote sinking air. Note that the longitudes plotted vary slightly in each of the four time panes to account for zonal movement. This analysis framework closely follows that originally outlined in Klemp et al. (2015).

All model solutions show bulk similarities. With respect to vertical velocity, a single, horseshoe-shaped updraft is noted at 30min in all models, although the degree to which the maximum updraft velocities are centered on the equator vary. A corresponding downdraft is located immediately to the east of the region of maximum positive vertical velocity. This downdraft is single-lobed (e.g., ACME-A) or double-lobed (e.g., GEM) in all simulations. Separation of the initial updraft occurs by 60min across all models, although variance begins to develop in the meridional deviation from the equator of the splitting supercell. Models such as NICAM, FV³, OLAM, and ICON all have larger and more distinct north-south spatial separation, while FVM, GEM, ACME-A, and TEMPEST show only a few degrees of latitude between updraft cores.

Structural differences also begin to emerge at 60min. For example, FVM, GEM, ACME-A, and TEMPEST all exhibit three local maxima in vertical velocity; two large updrafts mirrored about the equator with one small maximum still located over equator centered near the initial perturbation. Similar behavior is noted in the $q_r$ fields. This is in contrast with other models

which lack a third updraft on the equatorial plane. Generally speaking, $q_r$ maxima are collocated with the locations of maximum updraft velocities, and thereby conversion from $q_v$ and $q_c$ to $q_r$ in the Kessler microphysics.

While the aggregate response of a single updraft eventually splitting into poleward-propagating symmetric storms about the equator is well-matched between the configurations, notable differences exist, particularly towards the end of the runs. At 120min, FVM, GEM, ACME-A, OLAM, and MPAS all show two discrete supercells approximately $30°$ from the equator. FV[3] and TEMPEST both produce longitudinally-transverse storms that stretch towards the equator in addition to the two main cells. Each of the splitting supercells split a second time in ICON, forming, in conjunction with a local maximum at the equator, five maxima of vertical velocity (and correspondingly rainwater). NICAM produces two core supercells (as more clearly evident in the $q_r$ field at 120min), but has noticeable alternating weak updrafts and downdrafts in the north-south space between the two storm cores.

The relative smoothness of the storms as measured by the vertical velocity and rain water fields also varies between models, particularly at later times. ACME-A, FVM, GEM, OLAM, and MPAS produce updrafts that are relatively free of additional, small-scale local extrema in the vicinity of the core of the splitting supercell. Conversely, CSU, FV[3], ICON, NICAM, and TEMPEST all exhibit solutions with additional convective structures, with multiple updraft maxima versus two coherent cells. This spread is somewhat minimized when looking at rain water, implying that the overall dynamical character of the cells as noted by precipitation generation is more similar, with all models showing cohesive rain water maxima O(10 g/kg).

### 3.2 Resolution sensitivity of supercell

Fig. 4 shows the same cross-section variables as Fig. 3 except across the four specified test resolutions (nominally 4km, 2km, 1km, 0.5km, from left to right) at test termination of 120min. Therefore, the third panel from the left for each model (1km) should match the fourth panel from the left for each model in Fig. 3.

As resolution increases (left to right) models show increasing horizontal structure in both the vertical velocity and rain water fields. Updraft velocity generally increases with resolution, particularly going from 4km to 2km, implying that the supercell is underresolved at 4km resolution. This is supported by previous mesoscale simulations investigating supercells in other frameworks (Potvin and Flora, 2015; Schwartz et al., 2017), although it should be emphasized that this response is also subject to each numerical scheme's effective resolution (Skamarock, 2004) and that the resolvability of real-world supercells can depend on the size of individual storms.

At the highest resolutions, there is a distinct group of models that exhibit more small-scale structure, particularly in vertical velocity, at +120min at higher resolutions. CSU, GEM, and NICAM appear to have the largest vertical velocity variability at 0.5km, while ACME-A, FVM, MPAS, and TEMPEST appear to produce the smoothest solutions. This result is likely due to the differences in explicit diffusion treatment as noted before, as well as differences in the numerical schemes' implicit diffusion, particularly given the large impact of dissipation on kinetic energy near the grid scale (Skamarock, 2004; Jablonowski and Williamson, 2011; Guimond et al., 2016; Kühnlein et al., 2019). Additional focused sensitivity runs varying explicit diffusion operators and magnitude may be insightful for developers to explore. It is also hypothesized that differences in the coupling between the dynamical core and subgrid parameterizations may lead to some of these behaviors (e.g., Staniforth et al. (2002);

Gallus and Bresch (2006); Malardel (2010); Thatcher and Jablonowski (2016); Gross et al. (2018)) although more constrained simulations isolating physics-dynamics coupling in particular modeling frameworks is a target for future work. As before, rain water cross-sections tend to be less spatially variable at 0.5km than vertical velocity, although CSU and NICAM both show some additional local maxima in the field associated with some of the aforementioned $w$ maxima.

## 3.3 Convergence of global supercell quantities with resolution

While Fig. 4 highlights the structural convergence with resolution more storm-wide measures of supercell intensity are also of interest. Fig. 5 shows the maximum resolved updraft velocity over the global domain as a function of time for each dynamical core and each resolution (finer model resolution is denoted by progressively darker lines). Maximum updraft velocity is chosen as a metric of interest due to its common use in both observational and modeling studies of supercells. All models show increasing updraft velocity as a function of resolution, further confirming that, at 4km, the supercell is underresolved dynamically. For the majority of models and integration times, the gap between 4km and 2km grid spacing is the largest in magnitude, with subsequent increases in updraft velocity being smaller as models further decrease horizontal grid spacing. At 0.5km, the majority of models are relatively converged, with FV$^3$, ICON, and MPAS showing curves nearly on top of one another at these resolutions. Other models show larger differences between 0.5km and 1km curves, implying that these configuration may not yet be converged in this bulk sense. Further grid refinement or modifications to the dissipation schemes are necessary to achieve convergence; this is left to the individual modeling groups to verify (see Section 3.4 for an example).

The maximum updraft velocity as a function of resolution for particular model configurations varies quite widely. NICAM produces the weakest supercell, with velocities around 30 m s$^{-1}$ at 0.5km, while ACME-A, TEMPEST, GEM, and CSU all produce supercells that surpass 55 m s$^{-1}$ at some point during the supercell evolution. Models that have weaker supercells at 0.5km tend to also have weaker supercells at 4km (e.g., NICAM) while the same is true for stronger supercells (e.g., TEMPEST), likely due to configuration sensitivity. This agrees with the already discussed structural plots (Fig. 4) which demonstrated model solutions were generally converging with resolution on an intramodel basis but not necessarily across models.

Fig. 6 shows the same analysis except for area-integrated precipitation rate for each model and each resolution. Similar results are noted as above – with most models showing large spread at the coarsest resolutions, but general convergence in precipitation by 0.5km. All models produce the most precipitation at 120min with the 4km simulation. This is consistent with Klemp et al. (2015), who postulated this behavior is due to increased spatial extent of available $q_r$ to fall out of the column at these grid spacings, even though updraft velocities are weaker at coarser resolutions. Unlike maximum vertical velocity, the integrated precipitation rate does not monotonically increase with resolution for most models. At 120min, integrated rates at 0.5km range by approximately a factor of three or four, from a low of 50-70 x10$^5$ kg s$^{-1}$ (ACME-A, FVM, OLAM) to a high of 170-200 x10$^5$ kg s$^{-1}$ (CSU, FV$^3$), highlighting the sensitivity of final results that have already been discussed.

In addition to Figs. 5 and 6, which directly correspond to analysis in Klemp et al. (2015), we also define a storm-integrated kinetic energy ($IKE$) metric as follows:

$$IKE(t) = \frac{1}{2} \int\limits_0^{z_t} \int\limits_0^{A_e} \rho(u'^2 + v'^2 + w'^2)\,dA\,dz \tag{20}$$

where $z_t$ is the model top, $A_e$ is the area of the sphere, and winds $(u', v', w')$ are calculated as perturbations from the initial model state at the corresponding spatial location (e.g., $u' = u'(t, \phi, \psi, z) = u(t, \phi, \psi, z) - u(0, \phi, \psi, z)$). Here, local air density, $\rho(t, \phi, \psi, z)$, is computed using a standard atmosphere due to limitations in available data from some groups.

As a metric, $IKE$ is less sensitive to grid-scale velocities and is also a more holistic measure of storm-integrated intensity. This is shown in Fig. 7. Results are generally analogous to those in Fig. 6. This should be expected since total precipitation within a supercell is tied to the spatial extent and magnitude of the upward velocities that dominate the $IKE$ term. Convergence behavior between 1km and 0.5km appears similar for each model as noted earlier. The total spread across models at the end of the simulation for the 0.5km simulations is also similar to that seen in Fig. 6, demonstrating the large range in 'converged' solutions across models due to the various design choices discussed earlier.

### 3.4 Sample experiments at 0.25km grid spacing

While the formal supercell test case definition at DCMIP2016 specified 0.5km grid spacing as the finest resolution for groups to submit, it is clear that full convergence has not been reached for some of the modeling groups (e.g., Section 3.3). To confirm that the solution still converges further, two groups (FVM and TEMPEST) completed an exploratory set of simulations at 0.25km resolution. Fig. 8 shows the structural grid spacing convergence at 120min for the two models from 2km to 0.25km. Note that the left three panels for each model in Fig. 8 should match the corresponding three rightmost panels in Fig. 4. Fig. 9 shows FVM and TEMPEST IKE results, including the 0.25km simulations.

For TEMPEST and FVM, results indicate solution differences are markedly smaller between 0.5km and 0.25km than between 1km and 0.5km, implying the test is not fully converged at 0.5km for these models. Therefore, 0.25km may be a better target for a reference grid spacing going forward.

It is worth noting that the reference solution in Klemp et al. (2015) is indeed converged at 0.5km, as are some of the models in DCMIP2016. Given this, it is unclear whether the need to go beyond DCMIP2016 protocols for 'full' convergence is due to the test case definition itself or, rather, the implementation of the test case in particular models. This is left for subsequent analyses. However, given this result, it is recommended that groups applying this test case in the future continue to push beyond the four resolutions specified here if convergence is not readily apparent in either storm structure or bulk quantities at 0.5km.

### 4 Conclusions

Non-hydrostatic dynamics are required for accurate representation of supercells. The results from this test case show that clear differences and uncertainties exist in storm evolution when comparing identically initialized dynamical cores at similar

nominal grid resolutions. Intramodel convergence in bulk, integrated quantities appears to generally occur at approximately 0.5km grid spacing. However, intermodel differences are quite large even at these resolutions. For example, maximum updraft velocity within a storm between two models may vary by almost a factor of two even at the highest resolutions assessed at DCMIP2016.

5 Structural convergence is weaker than bulk integrated metrics. Two-dimensional horizontal cross-sections through the supercells at various times show that some models are well-converged between 1km and 0.5km, while results from other models imply that finer resolutions are needed to assess whether convergence will occur with a particular test case formulation and model configuration. Interestingly, in some cases maximum, bulk quantities converge faster than snapshots of cross-sections.

 We postulate that these differences and uncertainties likely stem from not only the numerical discretization and grid differ-10 ences outlined in Ullrich et al. (2017), but also from the form and implementation of filtering mechanisms (either implicit or explicit) specific to each modeling center. The simulation of supercells at these resolutions are particularly sensitive to numerical diffusion since damping of prognostic variables in global models is occurring at or near the scales required for resolvability of the storm. This is different from other DCMIP2016 tests (baroclinic wave and tropical cyclone), which produced dynamics that were less non-hydrostatic in nature and required resolvable scales well coarser than the grid cell level. Further, since 15 DCMIP2016 did not formally specify a particular physics-dynamics coupling strategy, it would not be surprising for particular design choices regarding how the dynamical core is coupled to subgrid parameterizations to also impact results.

 Given the lack of an analytic solution, we emphasize that the goal of this paper is not to define particular supercells as optimal answers. Rather, the main intention of this test at DCMIP2016 was to produce a verifiable database for models to use as an initial comparison point when evaluating non-hydrostatic numerics in dynamical cores. Pushing grid spacings to 0.25km 20 and beyond to formalize convergence would be a useful endeavour in future application of this test, either at the modeling center level or as part of future iterations of DCMIP. Variable-resolution or regionally-refined dynamical cores may reduce the burden of such simulations, making them more palatable for researchers with limited computing resources.

 We acknowledge that as groups continue to develop non-hydrostatic modeling techniques that small changes in the treatment of diffusion in the dynamical core will likely lead to changes in their results from DCMIP2016. We recommend modeling 25 centers developing or optimizing non-hydrostatic dynamical cores perform this test and compare their solutions to the baselines contained in this manuscript as a check of sanity relative to a large and diverse group of next-generation dynamical cores actively being developed within the atmospheric modeling community.

**Code availability**

Information on the availability of source code for the models featured in this paper can be found in Ullrich et al. (2017). For 30 this particular test, the initialization routine, microphysics code, and sample plotting scripts are available at http://dx.doi.org/ 10.5281/zenodo.1298671.

*Author contributions.* CMZ prepared the text and corresponding figures in this manuscript. Model data and notations about model-specific configurations were provided by the individual modeling groups. PAU assisted with formatting of the test case description in Section 2.

*Acknowledgements.* DCMIP2016 is sponsored by the National Center for Atmospheric Research Computational Information Systems Laboratory, the Department of Energy Office of Science (award no. DE-SC0016015), the National Science Foundation (award no. 1629819), the National Aeronautics and Space Administration (award no. NNX16AK51G), the National Oceanic and Atmospheric Administration Great Lakes Environmental Research Laboratory (award no. NA12OAR4320071), the Office of Naval Research and CU Boulder Research Computing. This work was made possible with support from our student and postdoctoral participants: Sabina Abba Omar, Scott Bachman, Amanda Back, Tobias Bauer, Vinicius Capistrano, Spencer Clark, Ross Dixon, Christopher Eldred, Robert Fajber, Jared Ferguson, Emily Foshee, Ariane Frassoni, Alexander Goldstein, Jorge Guerra, Chasity Henson, Adam Herrington, Tsung-Lin Hsieh, Dave Lee, Theodore Letcher, Weiwei Li, Laura Mazzaro, Maximo Menchaca, Jonathan Meyer, Farshid Nazari, John O'Brien, Bjarke Tobias Olsen, Hossein Parishani, Charles Pelletier, Thomas Rackow, Kabir Rasouli, Cameron Rencurrel, Koichi Sakaguchi, Gökhan Sever, James Shaw, Konrad Simon, Abhishekh Srivastava, Nicholas Szapiro, Kazushi Takemura, Pushp Raj Tiwari, Chii-Yun Tsai, Richard Urata, Karin van der Wiel, Lei Wang, Eric Wolf, Zheng Wu, Haiyang Yu, Sungduk Yu and Jiawei Zhuang. We would also like to thank Rich Loft, Cecilia Banner, Kathryn Peczkowicz and Rory Kelly (NCAR), Carmen Ho, Perla Dinger, and Gina Skyberg (UC Davis) and Kristi Hansen (University of Michigan) for administrative support during the workshop and summer school. The National Center for Atmospheric Research is sponsored by the National Science Foundation. Sandia National Laboratories is a multimission laboratory managed and operated by National Technology and Engineering Solutions of Sandia, LLC, a wholly owned subsidiary of Honeywell International Inc., for the U.S. Department of Energy's National Nuclear Security Administration under contract DE-NA0003525.

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

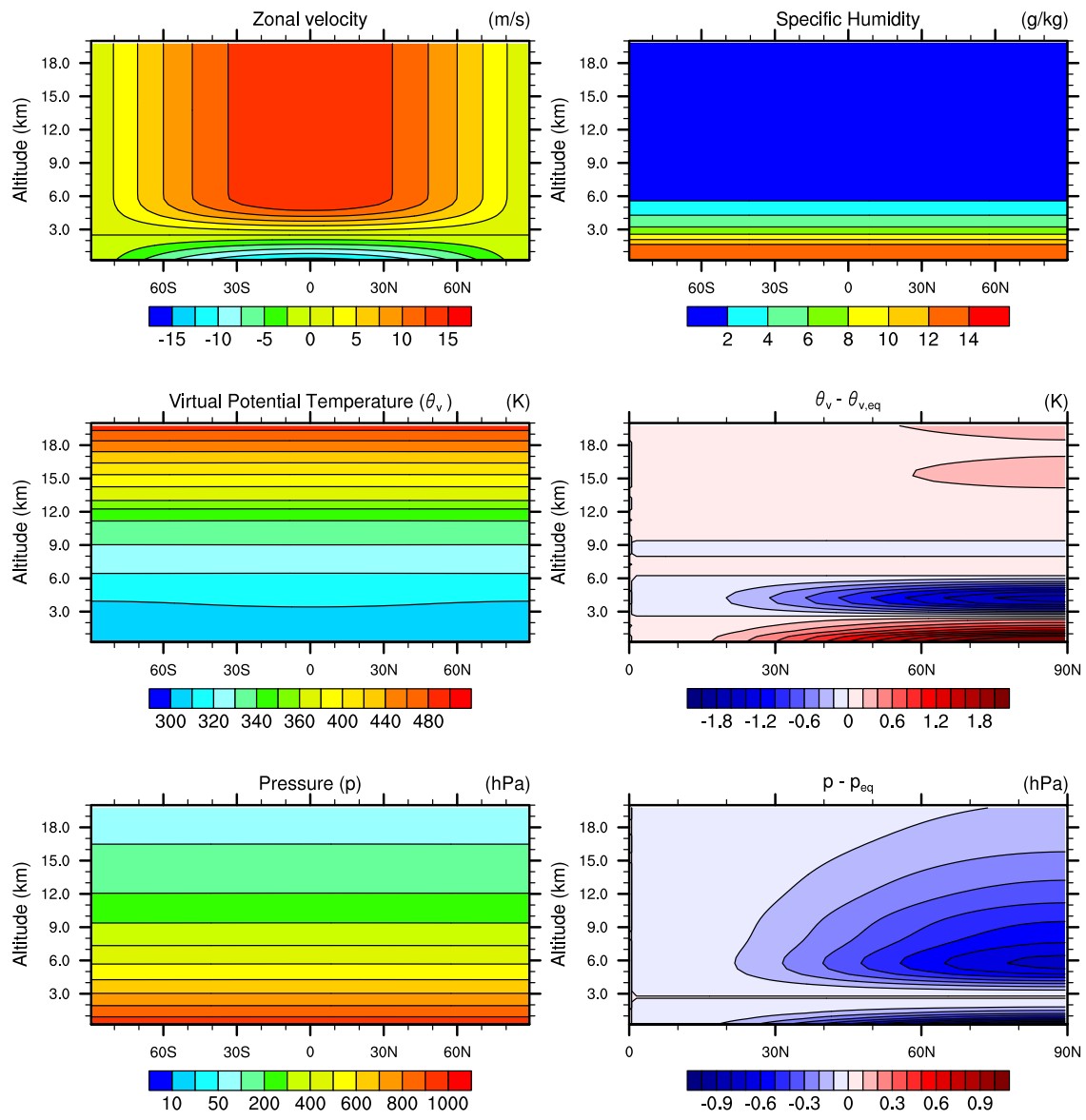

**Figure 1.** Initial state for the supercell test. All plots are latitude-height slices at $0°$ longitude. Deviations from equatorial values are shown for virtual potential temperature and pressure.

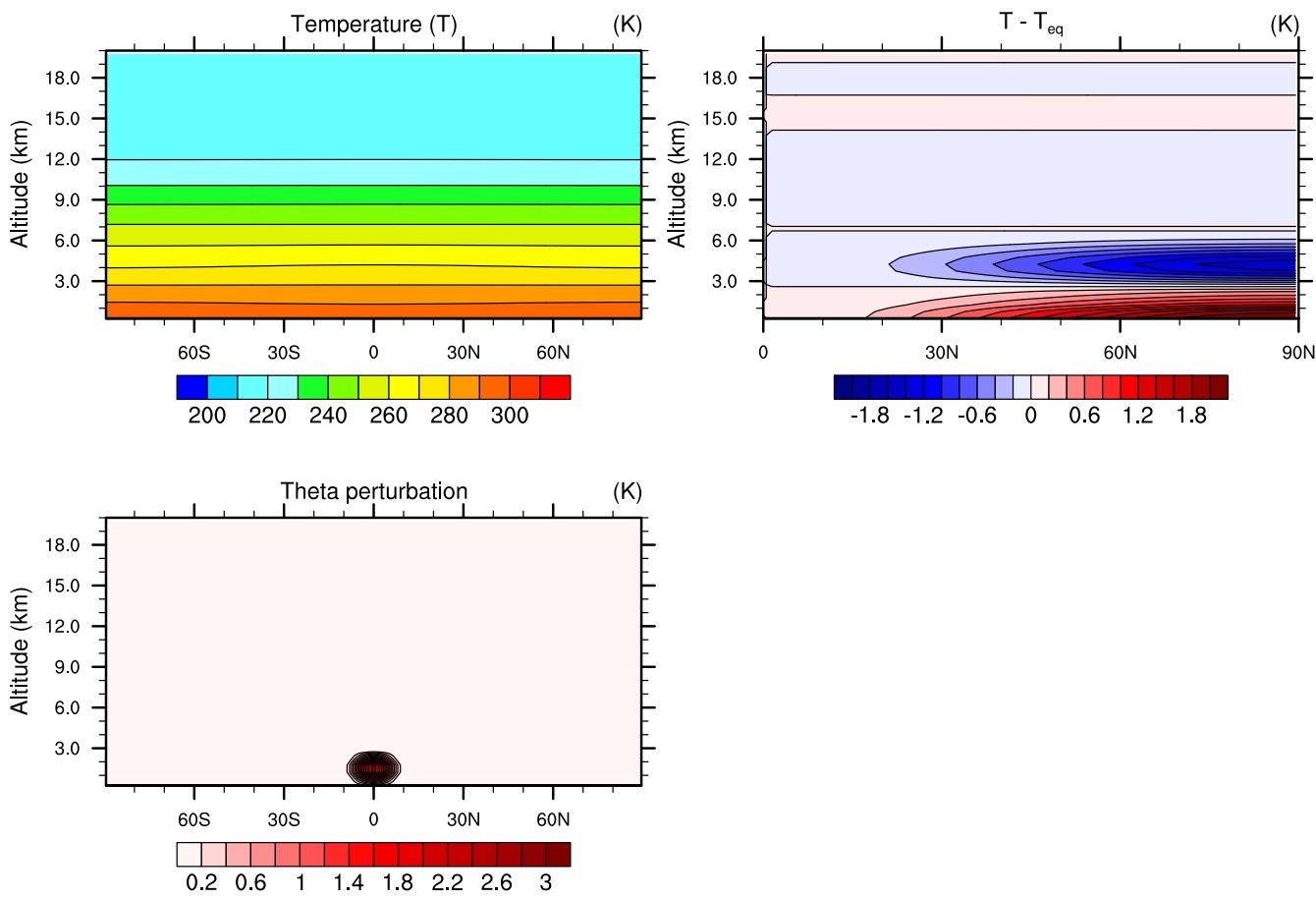

**Figure 2.** Same as Fig. 1 for temperature and potential temperature.

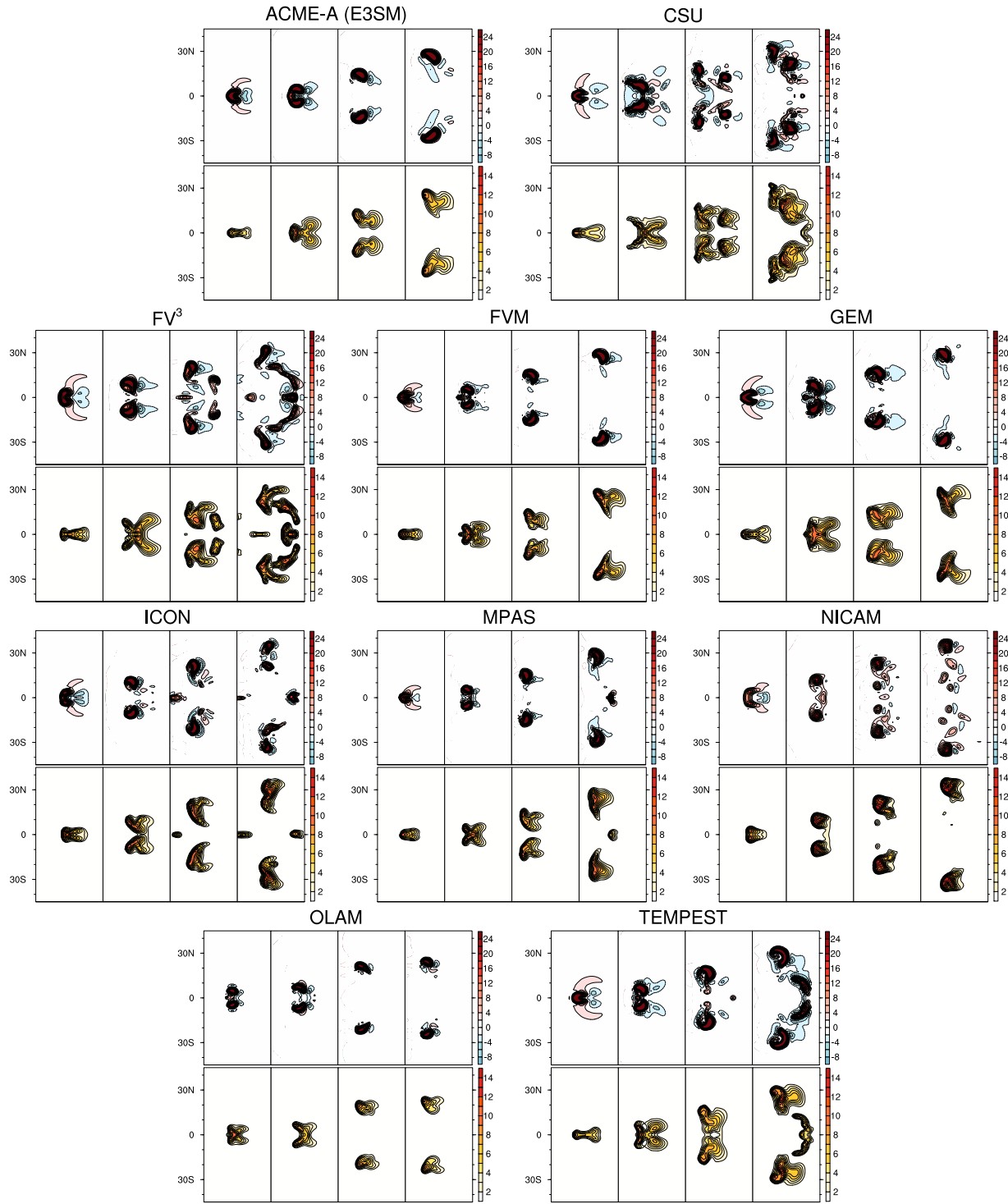

**Figure 3.** Time evolution of cross-sections of 5km vertical velocity (m s$^{-1}$, top) and 5km rain water (g kg$^{-1}$, bottom) for each model with the r100 configuration of the test case. From left to right, fields are plotted at 30min, 60min, 90min, and 120min.

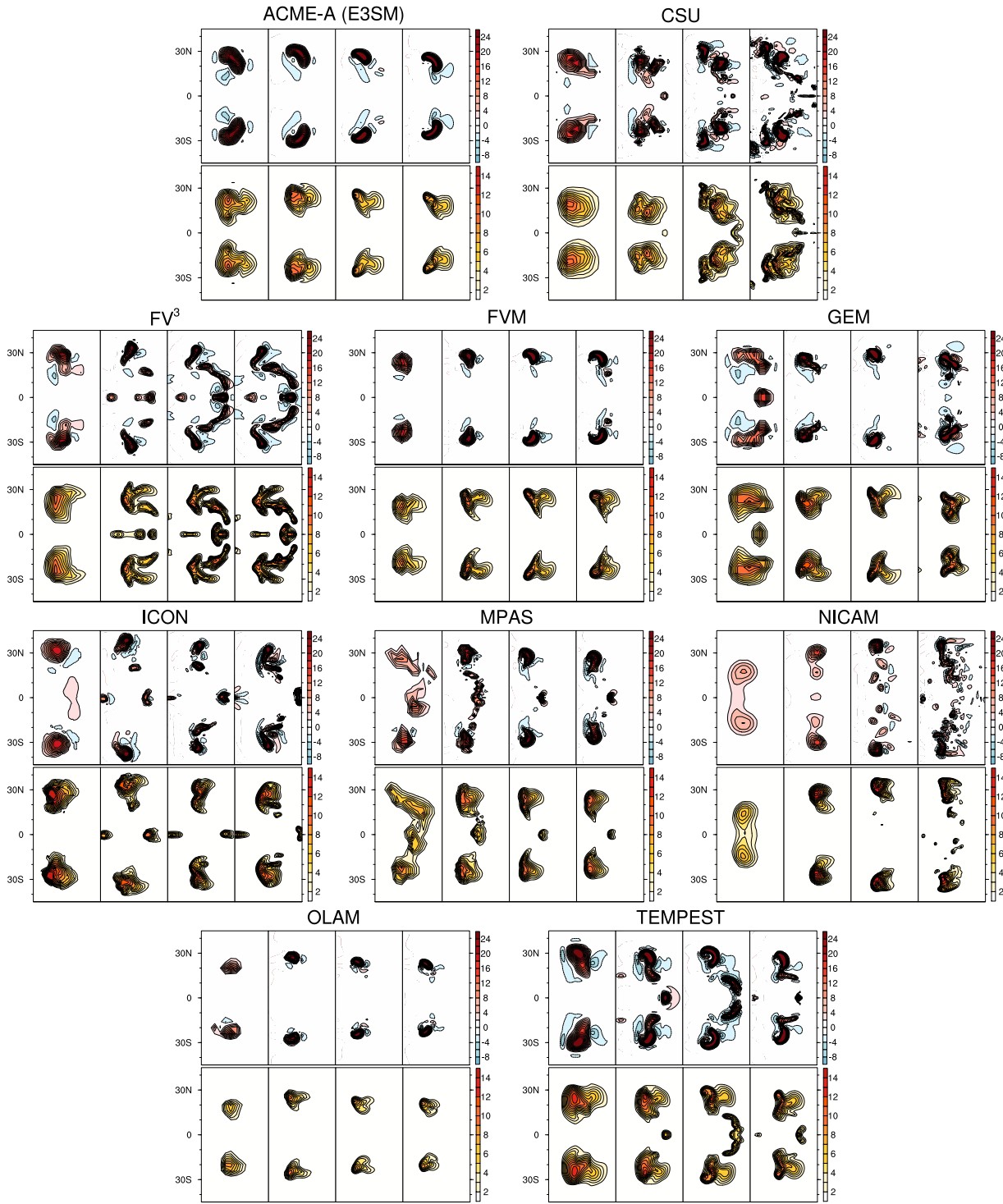

**Figure 4.** Resolution sensitivity of cross-sections of 5km vertical velocity (m s$^{-1}$, top) and 5km rain water (g kg$^{-1}$, bottom) plotted at 120min for each model. From left to right, nominal model resolutions are 4km, 2km, 1km, and 0.5km.

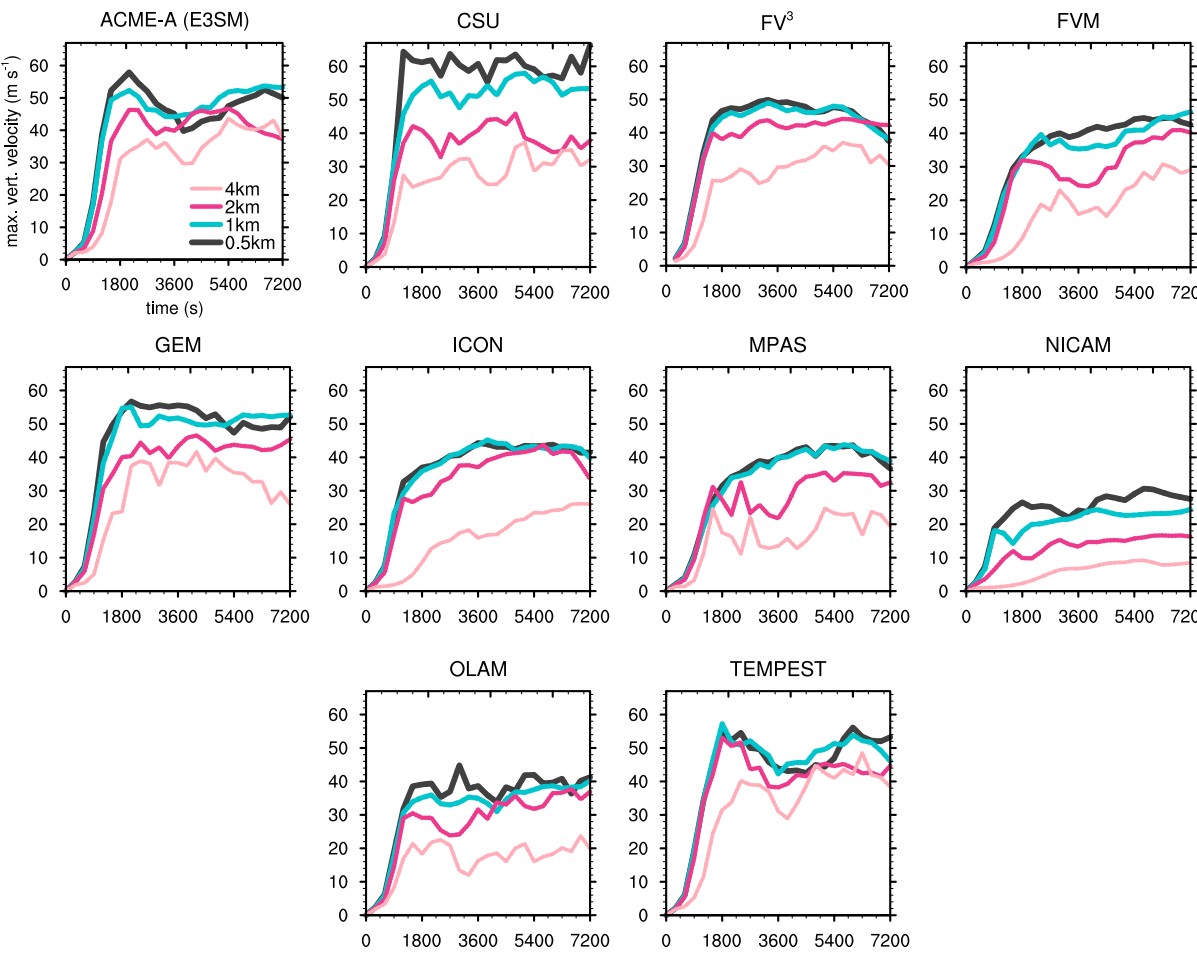

**Figure 5.** Maximum domain updraft velocity (m s$^{-1}$) as a function of time (seconds from initialization) for each model at each of the four specified resolutions. Note that the dark gray line is the finest grid spacing (0.5km) in this test.

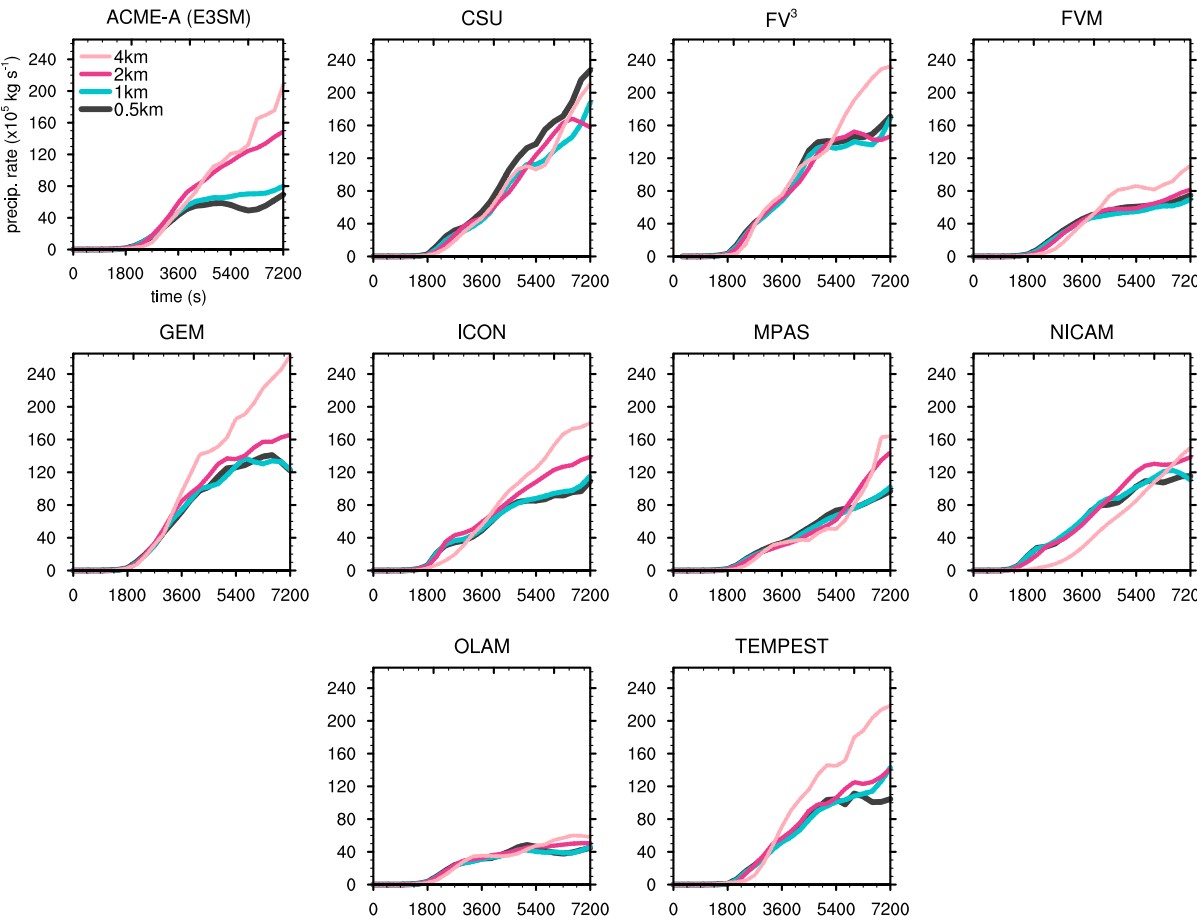

**Figure 6.** Same as Fig. 5 except showing area-integrated instantaneous precipitation rate $(\mathrm{x}10^5 \text{ kg s}^{-1})$.

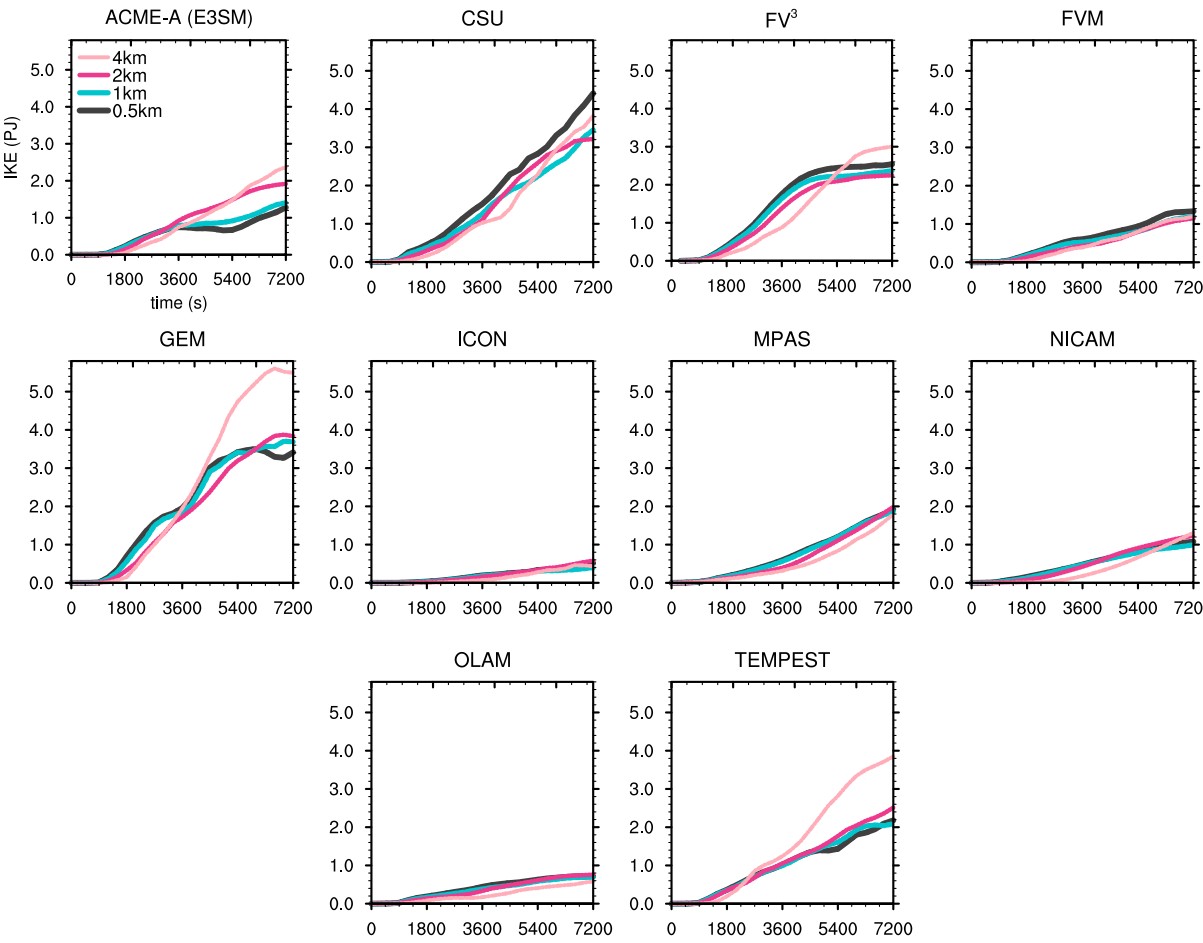

**Figure 7.** Same as Fig. 5 except showing storm-integrated kinetic energy (PJ) as defined in Eq. 20.

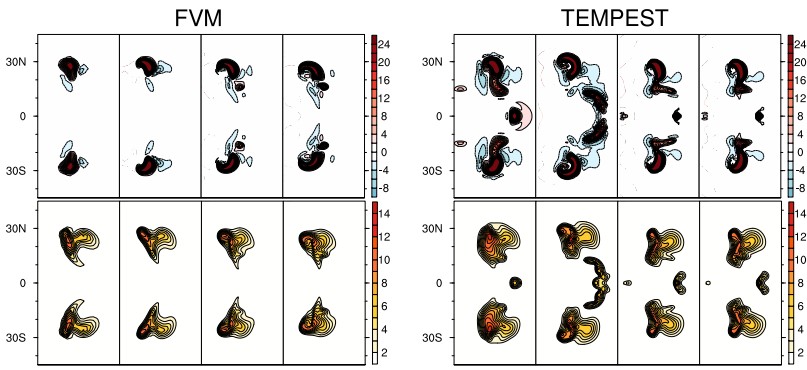

**Figure 8.** As in Fig. 4, except showing the subset of models that completed a 0.25km test. From left to right, nominal model resolutions are 2km, 1km, 0.5km, and 0.25km.

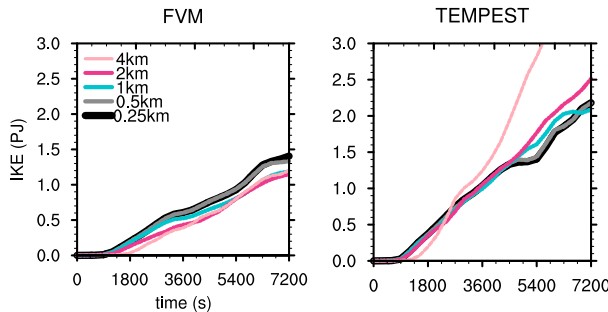

**Figure 9.** As in Fig. 7, except showing the subset of models that completed a 0.25km test.

**Table 1.** List of constants used for the Supercell test.

| Constant | Value | Description |
| --- | --- | --- |
| $X$ | 120 | Small-planet scaling factor (reduced Earth) |
| $\theta_{tr}$ | 343 K | Temperature at the tropopause |
| $\theta_0$ | 300 K | Temperature at the equatorial surface |
| $z_{tr}$ | 12000 m | Altitude of the tropopause |
| $T_{tr}$ | 213 K | Temperature at the tropopause |
| $U_s$ | 30 m/s | Wind shear velocity |
| $U_c$ | 15 m/s | Coordinate reference velocity |
| $z_s$ | 5000 m | Height of shear layer top |
| $\Delta z_u$ | 1000 m | Transition distance of velocity |
| $\Delta\theta$ | 3 K | Thermal perturbation magnitude |
| $\lambda_p$ | 0 | Thermal perturbation longitude |
| $\varphi_p$ | 0 | Thermal perturbation latitude |
| $r_p$ | $X \times 10000$ m | Perturbation horizontal half-width |
| $z_c$ | 1500 m | Perturbation center altitude |
| $z_p$ | 1500 m | Perturbation vertical half-width |

**Table 2.** Participating modeling centers and associated dynamical cores that submitted results for the splitting supercell test.

| Short Name | Long Name | Modeling Center or Group |
|---|---|---|
| ACME–A (E3SM) | Energy Exascale Earth System Model | Sandia National Laboratories and |
| | | University of Colorado, Boulder, USA |
| CSU | Colorado State University Model | Colorado State University, USA |
| FV$^3$ | GFDL Finite-Volume Cubed-Sphere Dynamical Core | Geophysical Fluid Dynamics Laboratory, USA |
| FVM | Finite Volume Module of the Integrated Forecasting System | European Centre for Medium-Range Weather Forecasts |
| GEM | Global Environmental Multiscale model | Environment and Climate Change Canada |
| ICON | ICOsahedral Non-hydrostatic model | Max-Planck-Institut für Meteorologie / DWD, Germany |
| MPAS | Model for Prediction Across Scales | National Center for Atmospheric Research, USA |
| NICAM | Non-hydrostatic Icosahedral Atmospheric Model | AORI / JAMSTEC / AICS, Japan |
| OLAM | Ocean Land Atmosphere Model | Duke University / University of Miami, USA |
| TEMPEST | Tempest Non-hydrostatic Atmospheric Model | University of California, Davis, USA |