# Peer review of "DCMIP2016: The Splitting Supercell Test Case"

_Geoscientific Model Development, 2018_

## Short Comment (SC1) · 27 Aug 2018

For reasons of predictability could you please provide the reference to the particular release of DCMIP2016 the manuscript is referring to. As outlined in https://www.geoscientific-model-development.net/about/manuscript_types.html the preferred mechanism of reference is a DOI which can easily be created from a GitHub release using for instance Zenodo, see https://guides.github.com/activities/citable-code/ for details. .

Lutz Gross GMD Executive Editor

---

## Author Comment (AC1) · 28 Aug 2018

The code specific to DCMIP2016 is permanently archived via DOI:10.5281/zenodo.1298671

Individual model code can be found in the DCMIP2016 overview manuscript via DOI:10.5194/gmd-10-4477-2017

These will be added to the manuscript upon future revision.

---

## Referee Comment (RC1) · Anonymous Referee #1 · 17 Sep 2018

**1   General comments**

The authors present initial conditions and results of an idealized dynamical core test.

The contribution is very welcome and the attention to detail appreciated. Too often are test case descriptions incomplete, which can result in difficulties if not the failure when trying to implement and or reproduce the results.

The Scientific reproducibility was rated as good. It would be excellent if the source datasets (NetCDF files) for the figures were uploaded to, for example, figshare, and cited with their DOI. We would strongly suggest the authors do this.

[Figure]

**2  Specific comments**

On page 2, lines 15 and 16: It is not clear that it is necessary to add the (1°) and (1km). Rather it would be clearer to write, for example, "... the resolution is $4°/X \approx 440\text{km}/X \approx 4\text{km}$.

On page 2, line 23: maybe rewrite the current "... were required with time between outputs required to be ..." as " were stored for post-processing with a frequency of every 15 min or higher."

Page 3, line 7, 8, and 9: I would suggest to rewrite equation 3 without the $\cos\varphi$ term and hence replace $\bar{u}(\varphi, z)$ with $u_{eq}(z)$ and add to line 4: $u(z,\varphi) = u_{eq}(z)\cos\varphi$. Remove any overbar on $U$ and $H$. It is not subsequently used, and in this particular case, it is the mean of a constant, and therefore potentially misleading as one might think that there is a zonal variation.

Page 5, line 27: Please add a summary of the results presented to prepare the reader and fill the void in between 3. and 3.1

Page 6, line 18: Maybe "... a notable difference exists through the end of the runs ..." would better read "... a notable difference exists towards the end of the runs ..."? "through" to me would indicate from beginning to end. "This spread in model solution..." potentially has an unclear precedent. Maybe "..., the models start to *diverge* towards the end of the run. This *divergence* ..." without the emphasis, of course, would be more explicit?

**3  Technical corrections**

Page 15, Figure 5: either replace the "increasing darker" with the actual color names or refer to the legend. I find it difficult to say which color is darker or lighter (except the

black and light pink, naturally).

On page 2, line 22 (and elsewhere in the document): There is no need to state that 120min=7200s. Just say that it is 120min.

Page 7, line 3: As noted above, no need to add the 7200 sec

**4   Conclusion**

Apart from these minor and mostly technical issues, we would strongly suggest to accept and congratulate the authors for the clear and detailed presentation.

---

## Referee Comment (RC2) · Anonymous Referee #2 · 28 Oct 2018

General comments:

The authors present an idealized test case (supercell storm) for global atmospheric models and briefly discuss the results from a large set of models that participated in a model comparison project (DCMIP2016). The premise of the study is good and useful. Demonstrating variability in the simulation of atmospheric phenomena due to differences in the dynamic core is an important topic and one that is often overlooked by the science community. The methods, analysis and even the figures are almost identical to a paper by Klemp et al. (2015), so the paper adds nothing new in this respect. The usefulness of the paper comes from the presentation of results from a large and diverse set of models. Despite this, the paper needs significant revisions to improve the presentation and additional analysis/simulations are needed to make the

results clearer.

The introduction was disappointing. No literature review is done on the impacts of dynamic cores on atmospheric problems such as extreme weather events. There are recently published papers on this topic that should be cited. In addition, more discussion is needed on the motivation for the study. Why do we need to compare dynamic cores? Do we expect significant differences and if so, do these differences explain some of the uncertainty observed in forecasts of extreme weather?

The results of the resolution sensitivity study don't appear to be converged at 0.5 km for several models. For example, there are significant differences in one or more fields going from 1 km to 0.5 km resolution for the following models: CSU, NICAM, ICON, GEM, TEMPEST. The authors should show a 0.25 km resolution simulation for one or two of these models to demonstrate better convergence properties. In addition, shouldn't we expect that as the grid spacing becomes very small, the model solutions and bulk statistics should be fairly similar across models? This is not the case with the current results so they are likely far from converged. Some discussion and additional example tests (as described above) are needed.

Specific comments:

Page 1, line 8; What is meant by "physics-dynamics coupling"?

Page 1, line 11; What is the difference between convective-permitting and convective-allowing? These mean the same thing to me.

Page 2, sentence starting with "It is based on the work of..."; Sentence doesn't read well and needs a re-write.

Page 2, line 15; Need few sentences on what a "reduced-radius sphere" is and what it intends to represent. Is the use of a reduced radius sphere a good approximation to true radius, global dynamics? Why do you need to simulate a supercell to study the performance of a global model? Wouldn't a global phenomenon (with non-hydrostatic

processes) be better suited for studying the performance of a global model? How many horizontal grid points are used in the reduced radius simulations?

Section 2.1; The notation in this section is confusing. What is Ueq a function of , z? If Ueq is the zonal velocity at the equator, then isn't u = Ueq? I also don't understand the transition to defining ubar after u. This whole section needs to be described better.

Page 3, line 1; This can't be gradient wind balance because no Coriolis force is shown in the equation.

Page 4, line 7; what is machine epsilon? Are you referring to machine precision?

Page 5, line 3; Why is the warm bubble hydrostatically balanced? The introduction and abstract highlight the importance of non-hydrostatic processes for testing global models, so this doesn't make sense. The vertical accelerations should be fairly small for this warm bubble relative to the supercell that is simulated with very large vertical velocities. The balance adjustment is a physical process that should be tested in the models.

Page 5, line 13; Should say "second-order diffusion operator with a constant sub-grid scale viscosity (value) is applied...". Are you using the same values in the horizontal and vertical dimensions? Is that appropriate for the grid cell aspect ratio?

Page 5, sentence starting with "ICON applied..."; This doesn't sound like a departure from the default described above. Also, is diffusion really applied to the mass continuity equation (rho) in ICON? This is not typically done.

Section 3.2; This whole section is very subjective with no analysis to back up any of the claims. In order to comment on the reasons for the differences in the model runs, some analysis is needed.

Page 7, line 11; How do you know that this is noise? This statement is subjective and no quantitative analysis is done to back up this claim.

Page 7, line 15; This is a very weak statement and should be removed. What is meant by "coupling between the dynamical core and subgrid parameterization" and why would this lead to these differences?

Page 7, line 21; This is not a bulk, integrated measure of supercell intensity. How about showing the domain integrated kinetic energy or total energy?

Technical corrections:

---

## Author Comment (AC2) · 10 Dec 2018

The authors present initial conditions and results of an idealized dynamical core test.

The contribution is very welcome and the attention to detail appreciated. Too often are test case descriptions incomplete, which can result in difficulties if not the failure when trying to implement and or reproduce the results.

The Scientific reproducibility was rated as good. It would be excellent if the source datasets (NetCDF files) for the figures were uploaded to, for example, figshare, and cited with their DOI. We would strongly suggest the authors do this.

Thank you for your comprehensive review.

In response to data being made available with a DOI, we should note that we (erroneously) omitted the formal public references for both the initialization code via Zenodo (DOI:10.5281/zenodo.1298671) and the model source code (DOI:10.5194/gmd-10-4477-2017) in the original submission. These have both been added to the body of the revised manuscript, which should allow for full reproducibility.

We hope we have addressed all of the comments satisfactorily below.

On page 2, lines 15 and 16: It is not clear that it is necessary to add the  $(1^{\circ})$  and (1km). Rather it would be clearer to write, for example, "... the resolution is  $4^{\circ}/X \sim 440 \text{km}/X \sim 440 \text{km}$ .

Per this suggestion, changed to 'Therefore, for a 1° mesh, the grid spacing of the reduced radius sphere is approximately  $1^{\circ}/X \sim 111 \text{km}/X \sim 111 \text{km}/120 \sim 1 \text{km}$  near the equator.'

On page 2, line 23: maybe rewrite the current "... were required with time between outputs required to be ..." as " were stored for post-processing with a frequency of every 15 min or higher."

Changed to '... to the microphysical routines were stored for post-processing with a frequency of every 15 min or finer.'

Page 3, line 7, 8, and 9: I would suggest to rewrite equation 3 without the  $\cos \phi$  term and hence replace  $u - (\phi, z)$  with ueq(z) and add to line 4:  $u(z, \phi) = ueq(z) \cos \phi$ . Remove any overbar on U and H. It is not subsequently used, and in this particular case, it is the mean of a constant, and therefore potentially misleading as one might think that there is a zonal variation.

We have implemented a 'hybrid' modification, which is a mix of the original formulation and the reviewer's suggestion. Specifically, we have chosen to remove the overbars in that our intent was to emphasize that these are mean background state values (that the warm bubble perturbation is placed upon), but the reviewer is correct to note that the notation may actually introduce more confusion than clarification.

However, we have chosen to leave the definition of wind velocity as is. This is done primarily for continuity with Klemp et al. [2015], as well as the published DCMIP2016 initialization code (DOI:10.5281/zenodo.1298671).

**Page 5, line 27: Please add a summary of the results presented to prepare the reader and fill the void in between 3. and 3.1**

We have added a paragraph which broadly summarizes the results to be presented in the following sections. We also chose to use this space to emphasize that attributing model spread to particular model design choices is beyond the scope of this manuscript; this paper's purpose is to catalog and define the set of model results from DCMIP2016 efforts to be used for future reference.

Page 6, line 18: Maybe "... a notable difference exists through the end of the runs ..." would better read "... a notable difference exists towards the end of the runs ..."? "through" to me would indicate from beginning to end. "This spread in model solution..." potentially has an unclear precedent. Maybe "..., the models start to diverge towards the end of the run. This divergence ..." without the emphasis, of course, would be more explicit?

We agree this passage was somewhat sloppy. This has been modified to read '... notable differences exist, particularly towards the end of the runs. This divergence can be seen as early as 30min in some cases but is most notable at the test conclusion.'

Page 15, Figure 5: either replace the "increasing darker" with the actual color names or refer to the legend. I find it difficult to say which color is darker or lighter (except the black and light pink, naturally).

This has been changed to only note that black is the finest resolution applied in this test. The reader is referred to the legend to match the rest of the lines to their associated grid spacing.

On page 2, line 22 (and elsewhere in the document): There is no need to state that 120min=7200s. Just say that it is 120min.

Corrected.

Page 7, line 3: As noted above, no need to add the 7200 sec Corrected.

Apart from these minor and mostly technical issues, we would strongly suggest to accept and congratulate the authors for the clear and detailed presentation.

Thank you.

**References**

J. B. Klemp, W. C. Skamarock, and S.-H. Park. Idealized global nonhydrostatic atmospheric test cases on a reduced-radius sphere. *Journal of Advances in Modeling Earth Systems*, 7(3):1155–1177, 2015.

---

## Author Comment (AC3) · 10 Dec 2018

**The authors present an idealized test case (supercell storm) for global atmospheric models and briefly discuss the results from a large set of models that participated in a model comparison project (DCMIP2016). The premise of the study is good and useful. Demonstrating variability in the simulation of atmospheric phenomena due to differences in the dynamic core is an important topic and one that is often overlooked by the science community. The methods, analysis and even the figures are almost identical to a paper by Klemp et al. (2015), so the paper adds nothing new in this respect. The usefulness of the paper comes from the presentation of results from a large and diverse set of models. Despite this, the paper needs significant revisions to improve the presentation and additional analysis/simulations are needed to make the results clearer.**

Thank you for your thorough review. We agree that the main thrust of this manuscript is to document a set of well-vetted solutions from many world-class dynamical cores. This not only provides a snapshot of model status at the time of DCMIP2016, but also acts as an inventory for which future modeling endeavours can compare their high-resolution solutions against; both as a sanity check as well as hypothesizing dynamical core impacts on non-hydrostatic phenomena.

**The introduction was disappointing. No literature review is done on the impacts of dynamic cores on atmospheric problems such as extreme weather events. There are recently published papers on this topic that should be cited. In addition, more discussion is needed on the motivation for the study. Why do we need to compare dynamic cores? Do we expect significant differences and if so, do these differences explain some of the uncertainty observed in forecasts of extreme weather?**

We have added some additional citations regarding recent work tying dynamical cores to extreme weather. That said, we would be happy to include discussion of additional literature that we may have overlooked.

We have also added some discussion of DCMIP's motivation and why it matters at 'the end of the pipeline' with respect to weather and climate simulations.

**The results of the resolution sensitivity study don't appear to be converged at 0.5 km for several models. For example, there are significant differences in one or more fields going from 1 km to 0.5 km resolution for the following models: CSU, NICAM, ICON, GEM, TEMPEST. The authors should show a 0.25 km resolution simulation for one or two of these models to demonstrate better convergence properties. In addition, shouldn't we expect that as the grid spacing becomes very small, the model solutions and bulk statistics should be fairly similar across models? This is not the case with the current results so they are likely far from converged. Some discussion and additional example tests (as described above) are needed.**

We would expect intramodel model solutions and bulk statistics to begin the converge at the higher resolutions in this study. That said, one of the obvious findings of DCMIP2016 is that non-hydrostatic phenomena at the resolutions tested here remain sensitive to numerical scheme, diffusion, and physics-dynamics coupling in a way that large-scale features such as baroclinic instabilities are not [Jablonowski et al., 2016].

While additional convergence tests may be enlightening, it is also computationally (and logistically) burdensome to complete new simulations at 250m resolution. That said, we agree that the reviewer's concerns are valid and have made care to note that these are targets for additional simulations either at the individual modeling center level or for future DCMIP projects.

**Page 1, line 8; What is meant by "physics-dynamics coupling"?**

This is the technical coupling between the dynamical core and representations of subgrid processes; commonly referred to as 'physics.' [Gross et al., 2016, 2018].

Gross et al. [2018] defines *physics-dynamics coupling* (PDC) as '... bringing together all the various discretized components to create a coherent model will be referred to here as physics–dynamics coupling. The term physics–dynamics coupling has evolved from the fact that the resolved fluid dynamics components are commonly known as the dynamical cores or simply "dynamics," and the physical parameterizations that represent the unresolved and underresolved processes and the nonfluid dynamical processes are collectively referred to as "physics."'

**Page 1, line 11; What is the difference between convective-permitting and convective-allowing? These mean the same thing to me.**

There is a bit of a 'gray' area in the definition of this by various modelers. Colloquially, a model 'permits' the simulation of a phenomenon if it can be discerned, regardless of whether it is under-resolved or not. A model 'allows' a phenomenon if it can be discerned and is credibly resolved. That said, we understand now where this can be confusing and isn't critical within the abstract so we have chosen to just include the term 'convective-allowing' especially since regimes pertaining to resolved convection in the atmosphere are a continuum and not cut discretely.

**Page 2, sentence starting with "It is based on the work of. . ."; Sentence doesn't read well and needs a re-write.**

As requested, this passage has been rewritten as follows 'This test is based on the work of Klemp and Wilhelmson [1978] and Klemp et al. [2015] and assesses the performance of global models at extremely high spatial resolution. It has recently been used in the development of next-generation numerical weather prediction systems [Ji and Toepfer, 2016].'

**Page 2, line 15; Need few sentences on what a "reduced-radius sphere" is and what it intends to represent. Is the use of a reduced radius sphere a good approximation to true radius, global dynamics? Why do you need to simulate a supercell to study the performance of a global model? Wouldn't a global phenomenon (with non-hydrostatic processes) be better suited for studying the performance of a global model? How many horizontal grid points are used in the reduced radius simulations?**

Using a reduced radius sphere allows for computationally-efficient simulations of O(1km) grid spacings in global models without modifying the numerical framework. Simulating supercells are important for non-hydrostatic development because A) the storms strongly stress non-hydrostatic numerics, B) they represent key atmospheric phenomena with high societal relevance, making them of importance to both weather and climate modelers, and C) they are currently unresolved in most global numerical modeling frameworks but that is projected to change over the coming decade or two.

We have added 'Reducing the model's planetary radius allows for fine grid spacing to be achieved without the added computational expense associated with adding grid cells a standard global mesh in order to achieve non-hydrostatic resolutions [Kuang et al., 2005]. Wedi and Smolarkiewicz [2009] provide a detailed overview of the reduced-radius framework for testing global models.' to the text to shed light on this approach.

**Section 2.1; The notation in this section is confusing. What is Ueq a function of , z? If Ueq is the zonal velocity at the equator, then isn't u = Ueq? I also don't understand the transition to defining ubar after u. This whole section needs to be described better.**

This notation has been modified per the suggestions of both Reviewers #1 and #2.

**Page 3, line 1; This can't be gradient wind balance because no Coriolis force is shown in the equation.**

Our apologies for this confusion. Technically, this is more akin to cyclostrophic balance. The text has been clarified to address this. The equation here can be derived from the gradient wind equation by setting the Coriolis term equal to zero and allowing a strong local pressure gradient force to be balanced by centrifugal force at an arbitrary latitude $\phi$.

**Page 4, line 7; what is machine epsilon? Are you referring to machine precision?**

They are quite similar but also technically distinct.

*Machine precision* is effectively the accuracy of the basic arithmetic operations.

*Machine epsilon* is the discrete distance between (for example) 1 and the next 'resolved' floating point number.

In this case, we use machine epsilon because we define convergence as the time when the 'distance' between iteration $n$ and $n+1$ is less than can be 'resolved' by the minimum gap between two floating point numbers.

**Page 5, line 3; Why is the warm bubble hydrostatically balanced? The introduction and abstract highlight the importance of non-hydrostatic processes for testing global models, so this doesn't make sense. The vertical accelerations should be fairly small for this warm bubble relative to the supercell that is simulated with very large vertical velocities. The balance adjustment is a physical process that should be tested in the models.**

The bubble is hydrostatically balanced in order to result in a smoothly-evolving solution at test case onset when the flow has not developed strong non-hydrostatic characteristics. Technically, there is no requirement that the bubble be balanced; however a less carefully-designed perturbation will result in gravity waves associated with flow adjustment in the first few timesteps and/or a less realistic supercell evolution. However, the long-term behavior of the solution is largely insensitive to whether or not the field is rebalanced.

**Page 5, line 13; Should say "second-order diffusion operator with a constant sub-grid scale viscosity (value) is applied…". Are you using the same values in the horizontal and vertical dimensions? Is that appropriate for the grid cell aspect ratio?**

Changed to 'as resolution is increased for a given model, a second-order diffusion operator with a constant viscosity (value) is applied to all momentum equations.'

The same value is used in the horizontal and vertical directions (unless specified in the relevant passage in the text). Since this diffusion is added to mimic turbulent dissipation within supercells, the choice of the same value for all three dimensions should be reasonable and is consistent with Klemp et al. [2015].

**Page 5, sentence starting with "ICON applied. . ."; This doesn't sound like a departure from the default described above. Also, is diffusion really applied to the mass continuity equation (rho) in ICON? This is not typically done.**

ICON departed from the prescribed test since it did not apply any diffusion in the vertical. The formal definition of the test case includes diffusion on the three-dimensional momentum equations (vertical diffusion is only applies to the background state perturbation).

The developers of ICON confirmed that the inclusion of $\rho$ in the list of variables where diffusion is applied was erroneous. This has been corrected and the developers greatly appreciate the reviewer noticing this oversight.

**Section 3.2; This whole section is very subjective with no analysis to back up any of the claims. In order to comment on the reasons for the differences in the model runs, some analysis is needed.**

We have added additional text to help buttress some of the hypotheses here. However, we should emphasize that the primary goal of this manuscript is not to do a formal deep dive into all model differences but rather define the test case and scope of solutions from modeling groups that participated in DCMIP2016. We have chosen to leave formal attribution studies to individual modeling centers (or groups of modeling centers) as model design choices are implemented when accounting for a host of considerations versus the outcome of a particular test case.

**Page 7, line 11; How do you know that this is noise? This statement is subjective and no quantitative analysis is done to back up this claim.**

This was poor verbiage on our part, we apologize. We have replaced 'noise' with 'small-scale structure' since we are not trying to argue these solutions necessarily contain numerical or physical 'noise;' spurious or otherwise.

**Page 7, line 15; This is a very weak statement and should be removed. What is meant by "coupling between the dynamical core and subgrid parameterization" and why would this lead to these differences?**

We have added text to clarify that this is a hypothesis and a target for future work. There has been previous work indicating that the coupling mechanisms between the dynamical core and subgrid physics parameterizations can drive sensitivity in solution output [Gross et al., 2018]. DCMIP2016 did not formally control for this, largely because it is very difficult to tightly specify a coupling framework that satisfies the multitude of different numerical schemes and software infrastructures used by various modeling centers. Rather, the majority of work investigating physics-dynamics coupling has been done within individual modeling frameworks, and is something a subset of participating DCMIP models will likely look into over the next few years.

**Page 7, line 21; This is not a bulk, integrated measure of supercell intensity. How about showing the domain integrated kinetic energy or total energy?**

For this test, integrated kinetic energy is a difficult value to evaluate, as the atmosphere is not at rest at initialization (and the storm dynamics' contribution to the kinetic energy budget is actually quite 'minimal'

relative to the KE of the background environment – on the order of 1% or so). Most of the energetic deviation from the initial sheared state is due to the strong vertical motion associated with the intensifying supercell.

We chose maximum updraft velocity over the entire domain as a metric for two reasons. One, it is a tangible quantity that is commonly reported in both observational and modeling studies of supercells. Two, we apply the assumption that maximum updraft velocity is a first-order proxy for 'storm intensity' as measured by a more 'dynamical' quantity than rainfall.

However, we note that the language here was not precise and understand why it may cause confusion, so we have modified to read '... more storm-wide measures...'

**References**

Markus Gross, Sylvie Malardel, Christiane Jablonowski, and Nigel Wood. Bridging the (knowledge) gap between physics and dynamics. *Bulletin of the American Meteorological Society*, 97(1):137–142, Jan 2016. doi: 10.1175/bams-d-15-00103.1. URL http://dx.doi.org/10.1175/BAMS-D-15-00103.1.

Markus Gross, Hui Wan, Philip J. Rasch, Peter M. Caldwell, David L. Williamson, Daniel Klocke, Christiane Jablonowski, Diana R. Thatcher, Nigel Wood, Mike Cullen, and et al. Physics–dynamics coupling in weather, climate, and earth system models: Challenges and recent progress. *Monthly Weather Review*, 146(11):3505–3544, Nov 2018. doi: 10.1175/mwr-d-17-0345.1. URL http://dx.doi.org/10.1175/MWR-D-17-0345.1.

C. Jablonowski, C. M. Zarzycki, K. A. Reed, P. A. Ullrich, J. Kent, P. H. Lauritzen, and R. D. Nair. The Dynamical Core Model Intercomparison Project (DCMIP-2016): Results of the Moist Baroclinic Wave Test Case. *AGU Fall Meeting Abstracts*, December 2016.

Ming Ji and Frederick Toepfer. Dynamical core evaluation test report for NOAA's Next Generation Global Prediction System (NGGPS). Technical report, National Oceanic and Atmospheric Administration, 2016. URL https://repository.library.noaa.gov/view/noaa/18653.

J. B. Klemp, W. C. Skamarock, and S.-H. Park. Idealized global nonhydrostatic atmospheric test cases on a reduced-radius sphere. *Journal of Advances in Modeling Earth Systems*, 7(3):1155–1177, 2015.

Joseph B Klemp and Robert B Wilhelmson. The simulation of three-dimensional convective storm dynamics. *Journal of the Atmospheric Sciences*, 35(6):1070–1096, 1978.

Zhiming Kuang, Peter N. Blossey, and Christopher S. Bretherton. A new approach for 3D cloud-resolving simulations of large-scale atmospheric circulation. *Geophysical Research Letters*, 32(2), 2005.

Nils P. Wedi and Piotr K. Smolarkiewicz. A framework for testing global non-hydrostatic models. *Quarterly Journal of the Royal Meteorological Society*, 135(639):469–484, 2009.

---

## Referee Report (RR1)

General comments:

I appreciate much of the authors work on the revision and the paper has improved. However, there are still a few items that were not addressed and the study still requires major revisions.

A simple google search of "impacts of dynamic cores on weather" reveals two studies on the impacts of dynamic cores on non-hydrostatic weather. These studies should be discussed in the paper as they are very relevant:

(1) Guimond et al. (2016), The impacts of dry dynamic cores on asymmetric hurricane intensification. This is a theoretical study that documents differences in hurricane intensification from inner-core asymmetries generated from heating (e.g. non-hydrostatic effects) due to dynamic cores. The authors analyze the physical and numerical reasons for these differences and is relevant to the present study.

Guimond, S.R., J.M. Reisner, S.M. Marras and F.X. Giraldo, 2016: The impacts of dry dynamic cores on asymmetric hurricane intensification. J. Atmos. Sci., 73, 4661 – 4684.

(2) Gallus and Bresch (2006), Comparison of impacts of WRF dynamic core, physics package, and initial conditions on warm season rainfall forecasts. This is an applied study that documents differences in the simulation of warm-season rainfall due to the choice of WRF dynamic core (either ARW or NMM) and physics package, among others. The interplay between the physics and dynamics is discussed and should be relevant to the present study although the grid spacing for this paper was coarse (~ 8 km).

Gallus, W.A., Jr., and J.F. Bresch, 2006: Comparison of impacts of WRF dynamic core, physics package, and initial conditions on warm season rainfall forecasts. Mon. Wea. Rev., 134, 2632 – 2641.

Regarding the convergence study, it would be too much to run 250 m simulations for all (or a group) of the models, but the request for a 250 m simulation for one or two of the identified models (CSU, NICAM, ICON, GEM, TEMPEST) shouldn't be asking too much given the large number of authors on the paper that are available for assistance. In addition, the simulations are for short times (120 min), which should help. I appreciate the authors work on this and I think it will help to address the spread observed in the models listed above.

Regarding the integrated measure of supercell intensity, I don't consider maximum vertical velocity a good choice because it is a point value that is highly sensitive to "noise" in the simulations and doesn't really reflect a bulk (or "storm-wide") measure of intensity. Figure 4 already showed snapshots of vertical velocity so we can already see the impacts on this field. Figure 6 shows area-integrated precipitation rate, which is a good global metric to show, but the authors need a global metric for kinetic or total energy in Figure 5. I don't see a problem calculating this: all models get the same initial conditions and supercell storms should be producing significant horizontal and vertical winds, you can focus in on the storm and integrate

over a smaller domain or remove a filtered field from the total fields to focus just on the storm perturbation energy.

Specific comments:

"This result is likely due to the differences in explicit diffusion treatment as noted before, as well as differences in the numerical schemes' implicit diffusion, particularly given the large impact of dissipation on kinetic energy near the grid scale (Skamarock, 2004; Jablonowski and 25 Williamson, 2011)."

--Reference (1) noted above (Guimond et al. 2016) also discuss the role of explicit and implicit diffusion in structural differences in non-hydrostatic weather and is relevant to the statement above.

"It is also hypothesized that differences in the coupling between the dynamical core and subgrid parameterizations may lead to some of these behaviors (e.g., Staniforth et al. (2002); Malardel (2010); Thatcher and Jablonowski (2016);"

--Reference (2) noted above (Gallus and Bresch 2006) also discuss this issue and is relevant to the statement above.

---

## Author Response (AR2)

**Second Response to Reviewer #2 of gmd-2018-156**

**I appreciate much of the authors work on the revision and the paper has improved. However, there are still a few items that were not addressed and the study still requires major revisions.**
We greatly appreciate the reviewer's continued review of this manuscript and believe many aspects of this analysis are now strengthened because of this feedback. Please see below for a point-by-point response and tracked changes.

**A simple google search of "impacts of dynamic cores on weather" reveals two studies on the impacts of dynamic cores on non-hydrostatic weather. These studies should be discussed in the paper as they are very relevant: Guimond et al. [2016] and Gallus and Bresch [2006].**
We thank the reviewer for bringing additional support for this work to light. The suggested references were not included in our original literature review because DCMIP2016 was explicitly focused on global dynamical cores. That said, upon further reflection, we recognize that these references are very much relevant to this specific case (the isolated role of numerics in simulation of extreme weather phenomena at non-hydrostatic scales), even with non-global domains, and have therefore been added to the manuscript.

**Regarding the convergence study, it would be too much to run 250 m simulations for all (or a group) of the models, but the request for a 250 m simulation for one or two of the identified models (CSU, NICAM, ICON, GEM, TEMPEST) shouldn't be asking too much given the large number of authors on the paper that are available for assistance. In addition, the simulations are for short times (120 min), which should help. I appreciate the authors work on this and I think it will help to address the spread observed in the models listed above.**
Following this review, two modeling groups (TEMPEST and FVM) agreed to produce 0.25km simulations for this manuscript. Both models show additional convergence from 0.5km to 0.25km; the fact that TEMPEST shows 0.25km results that are more similar to 0.5km than 0.5km is to 1km does imply the solution is not fully converged for all models at the finest grid spacing (0.5km) specified at DCMIP2016.
This has been noted in the manuscript through the addition of a new section (section 3.4). We also added specific language emphasizing that groups applying this test in the future may want to explore finer grid spacings, particularly if their solution does not appear converged at 0.5km.

**Regarding the integrated measure of supercell intensity, I don't consider maximum vertical velocity a good choice because it is a point value that is highly sensitive to "noise" in the simulations and doesn't really reflect a bulk (or "storm-wide") measure of intensity. Figure 4 already showed snapshots of vertical velocity so we can already see the impacts on this field. Figure 6 shows area-integrated precipitation rate, which is a good global metric to show, but the authors need a global metric for kinetic or total energy in Figure 5. I don't see a problem calculating this: all models get the same initial conditions and supercell storms should be producing significant horizontal and vertical winds, you can focus in on the storm and integrate over a smaller domain or remove a filtered field from the total fields to focus just on the storm perturbation energy.**
The reviewer's point regarding the potential sensitivity of maximum vertical velocity is well-taken. It is likely that this concern is only an issue in the event local grid cell vertical velocities unphysically exceed the general magnitude of those in the updraft, but we agree that a bulk value may capture systematic issues within the dynamical core.
We have defined the three-dimensional storm integrated kinetic energy ($IKE$) at a snapshot in time as follows:

$$IKE(t) = \frac{1}{2} \int_0^{z_t} \int_0^{A_e} \rho(u'^2 + v'^2 + w'^2) dA dz$$

where winds are calculated as perturbations from the initial state at the corresponding spatial location.
We have added this metric to the line convergence calculations as with integrated precipitation and maximum vertical velocity (both are retained to maintain consistency with the original Klemp et al. [2015] manuscript. Please see Fig. 7 (and Fig. 9) and corresponding discussion in the revised manuscript.

"This result is likely due to the differences in explicit diffusion treatment as noted before, as well as differences in the numerical schemes' implicit diffusion, particularly given the large impact of dissipation on kinetic energy near the grid scale (Skamarock, 2004; Jablonowski and 25 Williamson, 2011)." –Reference (1) noted above (Guimond et al. 2016) also discuss the role of explicit and implicit diffusion in structural differences in non-hydrostatic weather and is relevant to the statement above.

Thank you. This reference has been added accordingly.

"It is also hypothesized that differences in the coupling between the dynamical core and subgrid parameterizations may lead to some of these behaviors (e.g., Staniforth et al. (2002); Malardel (2010); Thatcher and Jablonowski (2016);" –Reference (2) noted above (Gallus and Bresch 2006) also discuss this issue and is relevant to the statement above.

This reference has also been added at this location.

**References**

[revised manuscript text omitted]